# Ist1 regulates ESCRT-III assembly and function during multivesicular endosome biogenesis in *Caenorhabditis elegans* embryos

E.B. Frankel[1], Raakhee Shankar[1], James J. Moresco[2], John R. Yates III [2], Niels Volkmann[3] & Anjon Audhya [1]

Degradation of most integral membrane proteins is directed by the endosomal sorting complex required for transport (ESCRT) machinery, which selectively targets ubiquitin-modified cargoes into intralumenal vesicles (ILVs) within multivesicular endosomes (MVEs). To better understand the mechanisms underlying ESCRT-mediated formation of ILVs, we exploited the rapid, de novo biogenesis of MVEs during the oocyte-to-embryo transition in *C. elegans.* In contrast to previous models suggesting that ILVs form individually, we demonstrate that they remain tethered to one another subsequent to internalization, arguing that they bud continuously from stable subdomains. In addition, we show that membrane bending and ILV formation are directed specifically by the ESCRT-III complex in vivo in a manner regulated by Ist1, which promotes ESCRT-III assembly and inhibits the incorporation of upstream ESCRT components into ILVs. Our findings underscore essential actions for ESCRT-III in membrane remodeling, cargo selection, and cargo retention, which act repetitively to maximize the rate of ILV formation.

---

[1] Department of Biomolecular Chemistry, University of Wisconsin-Madison School of Medicine and Public Health, 440 Henry Mall, Madison, WI 53706, USA. [2] The Scripps Research Institute, 10550 North Torrey Pines Rd., Department of Chemical Physiology, La Jolla, CA 92037, USA. [3] Bioinformatics and Structural Biology Program, Sanford-Burnham Medical Research Institute, 10901N Torrey Pines Rd, La Jolla, CA 92037, USA. Correspondence and requests for materials should be addressed to A.A. (email: audhya@wisc.edu)

The endosomal sorting complex required for transport (ESCRT) machinery plays key roles in several topologically equivalent membrane remodeling processes[1, 2]. Although initially described for its role in multivesicular endosome (MVE) biogenesis more than a decade ago[3–7], the mechanisms by which the ESCRT machinery functions to generate intralumenal vesicles (ILVs) harboring the appropriate cargoes remain unclear. A significant hurdle in the effort to characterize ESCRT action has been the lack of tractable systems to examine de novo MVE formation in the absence of specific ESCRT components. In yeast, none of the core ESCRT subunits are essential, but deletion of each results in the formation of morphologically aberrant endosomes at steady state, commonly referred to as Class 'E' compartments[8–10]. Such perturbations to endosome structure introduce major challenges to understand how individual ESCRT proteins contribute to cargo retention and/or membrane bending. In addition, constitutive loss of an ESCRT subunit may alter the expression or function of others, which could mask or suppress associated phenotypes. In mammalian cultured cells, acute depletion of ESCRT subunits avoids this issue, but requires multiple days of siRNA treatment, during which time pleiotropic effects on endosome morphology accumulate[11–13]. Alternatively, poor depletion can lead to misinterpretation of the roles various ESCRT components play during MVE formation. Reconstitution of ESCRT-mediated vesicle formation on synthetic liposomes has been achieved, suggesting the possibility that the 'early-acting' ESCRT-I and ESCRT-II complexes direct membrane bending, while ESCRT-III filaments promote membrane scission[14]. However, other in vitro studies argue that ESCRT-III is sufficient to drive both processes, functioning as spiral springs that deform membranes via their polymerization and driving scission upon relaxation[15–17]. Although significant advances in our understanding of ESCRT function have been achieved, the specific requirements for each ESCRT complex during vesicle formation and cargo sequestration in vivo remain unclear.

In general, there is broad consensus in the field that ESCRT-III filaments exhibit a preference for membranes of high negative curvature[15, 17–20], which are characteristic of the cytosolic face of nascent ILV bud necks. Conversely, recent evidence suggests that the ESCRT-III subunit Ist1, together with its binding partner CHMP1B, assembles onto positively curved membrane tubes in vitro[21] and functions at recycling endosomes in cultured cells[22]. In the absence of Ist1 function, endosomal tubulation becomes exaggerated[22], suggesting a requirement for Ist1 in their scission. In yeast, deletion of Ist1 does not impair MVE biogenesis, but synthetic genetic interactions with other ESCRT factors suggest that it positively regulates ILV formation[23–25]. Depletion of human Ist1 also does not disrupt the trafficking of known ubiquitin-modified ESCRT cargoes[26], but sensitive assays to easily measure ILV formation in mammalian cells are currently lacking. Overall, the role of Ist1 in endosomal protein sorting remains poorly understood.

Here, we take advantage of a unique developmental transition in the C. elegans germline, in which MVEs form rapidly during the oocyte-to-embryo transition de novo. By systematically depleting key ESCRT subunits prior to the first time they act in MVE biogenesis, we provide direct evidence that ESCRT-III plays a key role to initiate membrane bending in vivo. Moreover, our findings strongly suggest that ILVs bud continuously from subdomains on the limiting membrane of MVEs to rapidly internalize cargo. Strikingly, inhibition of Ist1 dramatically perturbs ESCRT-III assembly, which impairs cargo retention within endosomal subdomains and simultaneously permits upstream ESCRT complexes to be internalized aberrantly into the few ILVs that continue to form. Collectively, our studies highlight an important role for Ist1 in ESCRT-III function and normal MVE biogenesis.

## Results

**De novo MVE biogenesis during C. elegans zygotic development.** The organization of the C. elegans germline provides a unique environment to study membrane dynamics in response to various development cues (Fig. 1a). Upon oocyte fertilization and ovulation, the plasma membrane undergoes a dramatic transition that results in the downregulation and replacement of oocyte-specific factors with proteins necessary for embryo development. For example, the C. elegans LDL receptor Rme2 plays an essential role in cholesterol uptake in oocytes, but is dispensable in embryos once eggshell formation creates a barrier to the surrounding environment[27]. Accordingly, Rme2::GFP is internalized from the plasma membrane after ovulation and degraded rapidly (Supplementary Movie 1). In a similar manner, the trafficking of GFP-tagged caveolin-1 (GFP::Cav1) during the oocyte-to-embryo transition is highly stereotyped[28, 29], enabling a time resolved analysis of several transport pathways, including ESCRT-dependent protein sorting (Supplementary Movies 2 and 3). In oocytes, Cav1 accumulates on stable cortical granules, which fuse with the plasma membrane following fertilization[28, 30]. Subsequently, GFP::Cav1 fluorescence is quenched rapidly in one-cell stage embryos, presumably due to the ESCRT-dependent deposition of Cav1 into acidified endosomal compartments[31] (Supplementary Movie 2). Consistent with this idea, depletion of the core ESCRT-III subunit Vps32 leads to the aberrant accumulation of Cav1 in multicellular embryos, while its distribution in oocytes is unaffected (Supplementary Movie 3). These data suggest that the activity of the ESCRT machinery is particularly high in one-cell stage embryos.

To determine whether MVEs are even present in C. elegans oocytes, intact animals were high-pressure frozen, and thin sections were analyzed by electron microscopy. Numerous organelles could be resolved easily using this approach, including mitochondria and endoplasmic reticulum, but we were unable to identify MVEs in any proximal oocytes within the germline (Fig. 1a and Supplementary Fig. 1a). In contrast, we consistently observed numerous MVEs in early one-cell stage zygotes (Supplementary Fig. 1b). Together, these data strongly suggest that MVEs are produced de novo following oocyte fertilization and ovulation, in a manner independent of exogenous stimulation, providing an unprecedented platform to analyze the mechanisms underlying their native formation.

To determine the distribution of ESCRT components during early zygotic development, we imaged embryos in which Cav1 internalization had just been initiated (~23 min post ovulation). At this timepoint, ESCRT-0 (STAM) and ESCRT-I (Tsg101) were observed to co-localize in live cell imaging studies (Supplementary Fig. 1c, d). Using confocal and super resolution STED microscopy, we found that Cav1 accumulated on endosomes harboring ESCRT-0 and ESCRT-III (Vps32) directly adjacent to the cell cortex (Fig. 1b–d). Strikingly, we found that both ESCRT complexes were associated with subdomains on endosomes, while Cav1 was more uniformly distributed, likely due to ongoing deposition of the cargo onto the newly formed MVEs (Fig. 1c, d). We validated these findings using immunogold electron microscopy (Fig. 1e). Specifically, we found that Cav1 labeling occurred as individual particles or in clusters (two or more gold particles within 20 nm) with approximately equal frequency, while ESCRT-0 was primarily found in clusters ($n = 28$ MVEs analyzed, $83 \pm 4$ % of gold particles; mean $\pm$ S.E.M.) (Fig. 1f). Dual immunogold labeling also confirmed the presence of ESCRT-III on cortical MVEs, with Cav1 clearly associated with ILVs soon

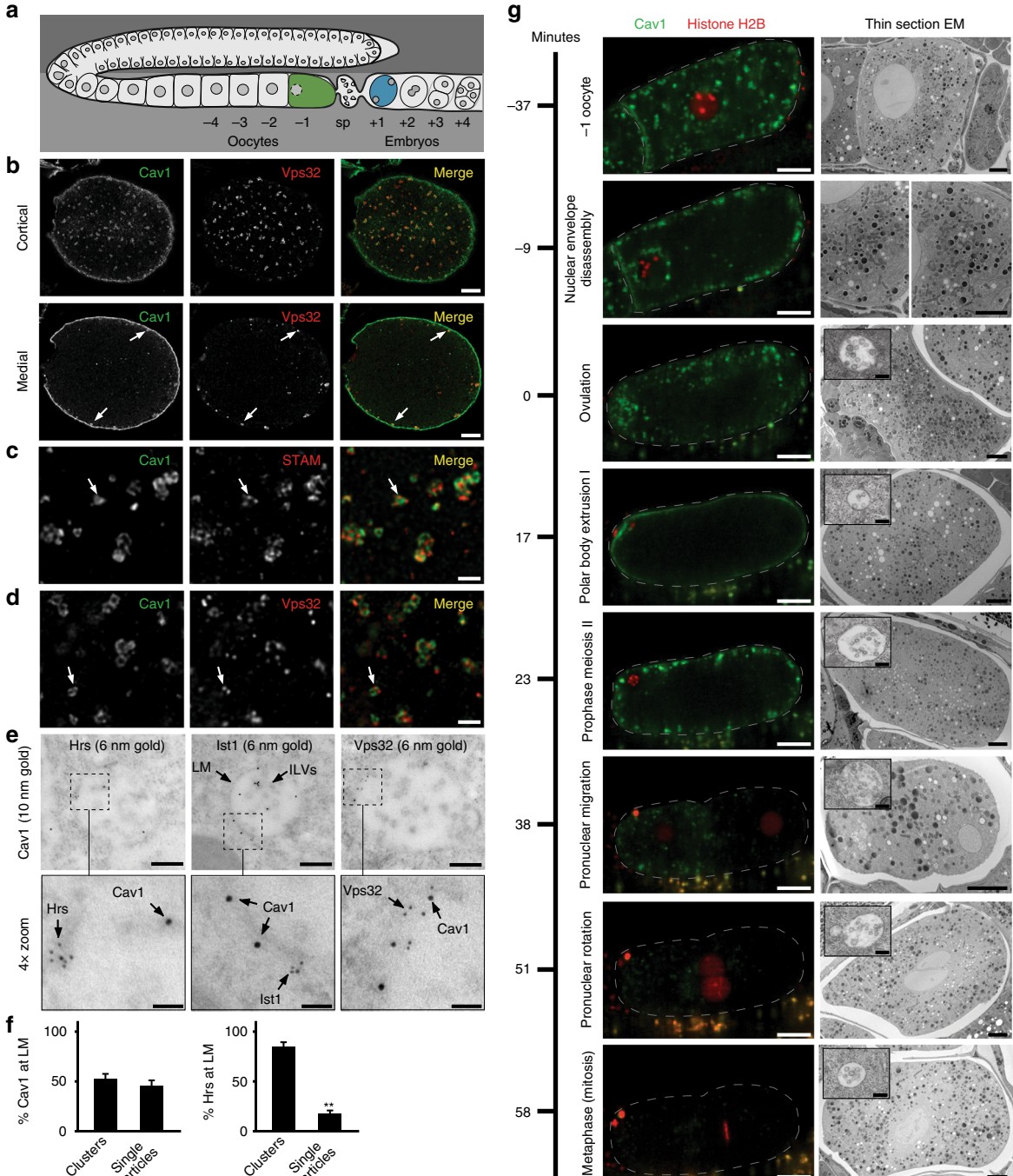

**Fig. 1** De novo MVE biogenesis initiates near the cortex of *C. elegans* zygotes. **a** Cartoon depicting the *C. elegans* reproductive system. Oocytes are fertilized as they pass through the spermatheca (sp) and develop as embryos within the uterus. **b** Embryos expressing GFP::Cav1 were fixed and stained using antibodies directed against GFP and Vps32 and imaged using confocal microscopy (*n* = 6). Representative images are shown. Arrows highlight cortical endosomes that are positive for both Cav1 and Vps32. Bars, 5 μm. **c, d** Higher magnification of embryos described in panel **b** imaged using STED microscopy. Arrows highlight the presence of endosomal subdomains harboring STAM (*n* = 5, **c**; ESCRT-0) or Vps32 (*n* = 5, panel **d**; ESCRT-III), while Cav1 is more uniformly distributed. Bars, 1 μm. **e** Animals were high-pressure frozen and processed for immunogold labeling using antibodies directed against Hrs (6 nm gold, left, *n* = 28 MVEs), Ist1 (6 nm gold, middle, *n* = 16 MVEs), or Vps32 (6 nm gold, right, *n* = 11 MVEs), together with antibodies against GFP:: Cav1 (10 nm gold). The limiting membrane (LM) and intralumenal vesicles (ILVs) are indicated with arrows (top, middle panel). Representative images are shown. Bars, 200 nm (top panels) and 50 nm (bottom panels). **f** Based on the distribution of gold labeling at the endosome limiting membrane (LM) described in panel **e**, the percentages of Cav1 and Hrs in clusters/subdomains (defined as 2 or more gold particles within 20 nm) or individual gold particles was quantified. Standard error of the mean is shown in each case. **p < 0.01, using a *t*-test. **g** Animals expressing GFP::Cav1 and RFP::His58 (Histone H2B) were imaged using time lapse confocal microscopy during ovulation and early embryogenesis (left panels; oocyte and embryo are outlined, *n* = 12 animals). Corresponding thin-section electron microscopy studies were conducted at the indicated time intervals (right panels). Representative images are shown. Bars, 10 μm (left panels), 4 μm (right panels), and 200 nm (right panel insets highlighting MVEs)

after its internalization from the plasma membrane (Fig. 1e). Together, these data indicate that ESCRT-mediated protein sorting initiates rapidly following cargo endocytosis, directly adjacent to the plasma membrane.

We next analyzed MVE morphology during different stages of zygotic development (Fig. 1g). These studies revealed oscillations in MVE diameter, which correlated with the timing of two endocytic waves that reproducibly occur upon embryo ovulation (Fig. 1g and Supplementary Fig. 1e). The first wave takes place coincidently with ovulation[32], and we found that the average diameter of MVEs (and the average number of ILVs within them) in zygotes increased steadily from this timepoint until completion of meiosis I and the first polar body extrusion event (Fig. 1g and Supplementary Fig. 1e, f). At this stage of development, there was a substantial decline in average MVE size and a corresponding decrease in ILV number. Immediately after meiosis I completes, a second wave of endocytosis occurs[32], and we again observed a steady increase in MVE diameter and ILV number until the maternal and paternal pronuclei were positioned at the center of the embryo. At this point, the average size of MVEs declined again, with fewer ILVs within them, and MVE morphology was subsequently maintained through mitosis (Fig. 1g and Supplementary Fig. 1e, f). By the two-cell stage, fewer MVEs in total were observed, but their average size and number of ILVs remained similar to those seen during the first embryonic mitosis (Fig. 1g and Supplementary Fig. 1e, f). Altogether, these data suggest that each wave of endocytosis contributes membrane to enlarging MVEs, until equilibrium is reached between ILV production, which consumes the limiting membrane of MVEs, and new membrane addition from other compartments.

**ILVs remain physically connected following internalization**. Based largely on the steady state distribution of ILVs within pre-existing MVEs, current models suggest that the ESCRT machinery assembles transiently to enable individual budding events, followed by Vps4-mediated disassembly[33–35]. However, using thin-section EM in early-stage zygotes, we found that ILVs often appeared to be in contact with one another inside newly formed MVEs, both near the limiting membrane and at their center (Fig. 2a). We used EM tomography to examine the distribution of ILVs in three dimensions, which demonstrated without the ambiguity of two-dimensional projection data that more than 80% were in direct contact with another ILV during early zygotic development (Table 1). These data suggest that ILV internalization events may be coupled to one another.

To explore this possibility further, we collected EM tomograms of newly synthesized MVEs in zygotes immediately following ovulation. At this early timepoint, ILVs appeared in clusters adjacent to the limiting membrane, often associated with an active budding site, as opposed to being randomly distributed within the endosome lumen (Fig. 2b). When we modeled the inner membranes of ILVs, we found that each was juxtaposed closely against a neighboring vesicle (Fig. 2b–d; ~ 15 nm, as measured between the inner membrane of one ILV to the inner membrane of another). We next modeled their outer membranes and found that ILVs were physically connected via electron dense tethers of varying width ($18 \pm 7$ nm; mean $\pm$ S.E.M., $n = 53$) and length ($12 \pm 5$ nm; mean $\pm$ S.E.M., $n = 53$) (Fig. 2e). Altogether, our findings are consistent with the idea that during rapid MVE biogenesis, ESCRT-mediated ILV budding occurs repetitively from stable endosomal subdomains, rather than as individual events.

To determine whether repetitive ILV formation in the absence of intermittent scission may be a more universal feature of ILV biogenesis and not limited to the oocyte-to-embryo transition, we examined hypodermal cells, in which relatively few MVEs are observed at steady state[36], similar to other model systems studied previously[34]. Using electron tomography, we found that the average diameter of ILVs in the hypodermis ($57.6 \pm 1$ nm; mean $\pm$ S.E.M., $n = 140$) was nearly identical to that found in one-cell stage embryos (Fig. 2f), and the majority ($88 \pm 3$ %; mean $\pm$ S.E.M., $n = 152$ ILVs) exhibited contact with another ILV. Moreover, by modeling the outer membranes of ILVs in the hypodermis, we again identified tethers of varying width ($16 \pm 5$ nm; mean $\pm$ S.E.M., $n = 54$) and length ($11 \pm 3$ nm; mean $\pm$ S.E.M., $n = 54$) that linked them together (Fig. 2g). These data strongly suggest that ILV formation events are coupled irrespective of flux through the ESCRT-mediated MVE biogenesis pathway.

**ESCRT-III promotes cargo clustering in vivo**. Our unique ability to examine de novo MVE biogenesis enables us to address the long-standing question of which ESCRT complexes direct cargo sequestration and membrane bending on endosomes in vivo. Contradictory studies have suggested that either early-acting or late-acting ESCRT components participate directly in these processes[14, 16]. To define the contributions of each, we conducted a series of RNAi-mediated depletion studies in *C. elegans* using both light and electron microscopy, focusing first on core subunits of the ESCRT-I (Tsg101) and ESCRT-III (Vps20 and Vps32) complexes. The efficacy of depletion was assayed by immunoblot analysis, immunofluorescence, and examining GFP:: Cav1 fluorescence (Fig. 3a and Supplementary Fig. 2a, b), which is normally extinguished rapidly after embryo fertilization, but persists in the absence of ESCRT function[29, 31]. In all ESCRT subunit depletion studies conducted, GFP::Cav1 remained visible by confocal imaging in embryos well beyond the two-cell stage (Fig. 3a and Supplementary Fig. 2a, c). Analysis of each condition by thin-section electron microscopy and electron tomography demonstrated an increase in the number of enlarged endosomes (>1 μm in diameter), although this population remained in the minority (Fig. 3b). The number of small endosomes (120–250 nm in diameter) that harbored Cav1 was also elevated in embryos depleted of ESCRT components, while the population of endosomes ranging in size from 250–600 nm in diameter was diminished (Fig. 3a, b and Supplementary Fig. 2d).

We also analyzed the distribution of Cav1 and Hrs (ESCRT-0) at the limiting membrane of endosomes using immunogold electron microscopy, specifically focusing on the impact of ESCRT-III inhibition (Fig. 3c, d and Supplementary Fig. 2e). Depletion of Vps32 resulted in the assembly of enlarged ESCRT-0 subdomains ($60.8 \pm 5$ nm in diameter; mean $\pm$ S.E.M., $n = 91$) relative to control samples ($37.5 \pm 3$ nm in diameter; mean $\pm$ S.E. M., $n = 60$) (Fig. 3d). However, we failed to observe a commensurate increase in Cav1 clustering under this condition (Fig. 3e). Moreover, depletion of the ESCRT-III subunit Ist1 resulted in a dramatic reduction in Cav1 cluster formation as compared to controls (Fig. 3e). Altogether, these data suggest that proper ESCRT-III assembly is required to maintain cargo clustering, likely in a manner that is independent of ESCRT-0 subdomain formation. Importantly, penetrant depletion of Vps32 potently blocked membrane bending on endosomes, indicating that ESCRT-III activity is required to initiate ILV formation in vivo (Figs. 3a, c, d). Although upstream ESCRT complexes are required to recruit and promote the assembly of ESCRT-III at MVEs, our data contrast prior in vitro studies, which argued that ESCRT-I and ESCRT-II specifically control the process of membrane bending[14].

We additionally explored the contribution of the ATPase Vps4 to ILV formation. Since penetrant depletion of Vps4 causes sterility in *C. elegans*[37], we investigated the impact of expressing a

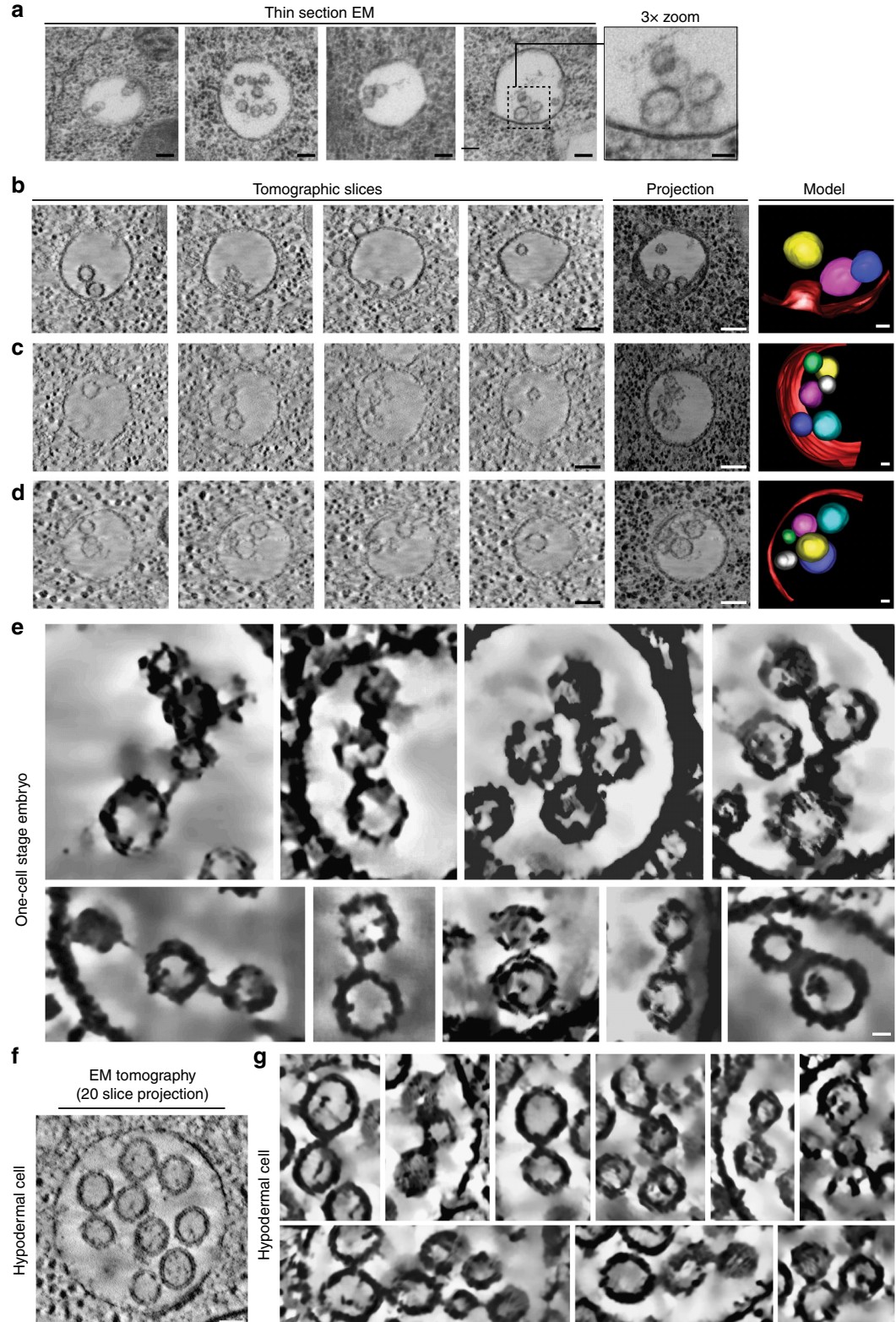

**Fig. 2** ILV formation occurs repetitively from endosomal subdomains. **a** High-pressure frozen animals were processed for thin-section electron microscopy to visualize newly formed MVEs within the one-cell stage embryo. Micrographs are representative of 51 MVEs visualized. Bars, 100 nm (left panels) and 50 nm (right, zoomed panel). **b**–**d** Several representative MVEs in newly formed zygotes were imaged using electron tomography. Individual slices and minimal intensity projections are shown (Bars, 100 nm), together with reconstructed models (right; Bars, 25 nm). **e** Regions extracted from several tomograms (from one-cell stage embryos) were processed for noise reduction and feature enhancement. Bar, 25 nm. **f** Representative MVE found in a control hypodermal cell imaged using electron tomography (*n* = 7). Bar, 50 nm. **g** Regions extracted from several tomograms (from hypodermal cells) were processed for noise reduction and feature enhancement. Bar, 25 nm

| **Table 1 Tomogram statistics for one-cell stage embryos** | |
| --- | --- |
| Intralumenal vesicle (ILV) diameter | 57 ($\pm$1) nm, $n = 534$ |
| Multivesicular endosome diameter | 401 ($\pm$12) nm, $n = 129$ |
| Percentage of ILVs in contact | 81 ($\pm$1)%, $n = 532$ |

The average size and distribution of ILVs and the diameter of wild type MVEs was determined using electron tomography. S.E.M. is reported in each case

dominant negative isoform harboring a well-characterized Walker B mutation that inhibits ATP hydrolysis[38]. For these studies, we generated animals expressing Vps4[E219Q] specifically in the germline, which resulted in 98.4% embryo lethality. Electron tomography of endosomes from one-cell stage embryos expressing Vps4[E219Q] revealed an increase in the population of enlarged endosomes (Fig. 3b and Supplementary Fig. 2f) and the presence of closely juxtaposed, nascent ILVs that remained connected to the limiting membrane (Supplementary Fig. 2f, g). On average, the minimum diameter of vesicle bud necks observed was ~20 nm, similar to the inner diameter of purified Vps32 spiral filaments[39]. These data are consistent with the idea that Vps4 ATPase activity plays an essential role in remodeling ESCRT-III filaments to facilitate completion of ILV budding[39].

Since full inhibition of ESCRT complexes blocked ILV formation, we conducted a series of partial depletions followed by electron microscopy to examine their individual contributions to this process. Under these conditions, we observed the formation of aberrant ILVs, which mostly remained clustered near the limiting membrane (Fig. 3a). Using electron tomography, we measured the diameters of ILVs formed, revealing that inhibition of both early-acting and late-acting core ESCRT components caused an increase in ILV size relative to controls (Table 2 and Supplementary Movies 4–8). These data are consistent with previous work demonstrating that perturbations to ESCRT-III recruitment and assembly can lead to the formation of abnormally large ILVs[34, 35, 40, 41].

**Ist1 regulates ESCRT-III assembly and function at MVEs**. The rapid rate of MVE biogenesis at the oocyte-to-embryo transition provides an ideal setting to revisit the role of Ist1, which was shown previously to associate with Vps4 and CHMP1 in human cells[23, 24, 42], but implicated more recently in facilitating membrane recycling at endosomes[22]. Analysis of Cav1 trafficking in embryos either depleted of Ist1 or harboring a mutation (tm7401) that truncates the last 84 amino acids and destabilizes the protein revealed a dramatic reduction in cargo degradation (Fig. 4a, b and Supplementary Fig. 3a, b). Analysis by electron tomography confirmed a potent defect in MVE biogenesis, highlighted by an accumulation of ILVs near the limiting membrane that continued to be tethered to one another (Fig. 4c). Although the tethers exhibited an indistinguishable length distribution as compared to controls (11 $\pm$ 5 nm; mean $\pm$ S.E.M., $n = 37$), their widths were significantly thinner (12 $\pm$ 5 nm; mean $\pm$ S.E.M., $n = 37$; $p < 0.0001$ using a $t$-test). In addition, in contrast to other ESCRT depletions, ILV diameter was significantly reduced following Ist1 inhibition, suggesting a direct role for Ist1 in uniquely regulating ESCRT-III assembly and/or function (Table 2).

These findings differ dramatically from those of several other groups, which previously failed to identify defects in ESCRT-mediated cargo degradation following Ist1 inhibition[23, 24, 26, 43, 44]. A major distinction between our respective approaches rests with our unique ability to analyze de novo MVE formation during a native period of high endocytic flux, while others have relied largely upon examination of cargo sorting at steady state. To define the impact of Ist1 inhibition in a more comparable

setting to that analyzed previously using other models, we visualized MVEs in hypodermal cells of ist1 (tm7401) mutant animals using electron tomography. These studies demonstrated that impaired Ist1 function does not cause an accumulation of ILVs near the limiting membrane when endocytic flux is low (Fig. 4d). However, we found that the average diameter of ILVs produced under this condition was still significantly reduced (45.7 $\pm$ 1 nm; mean $\pm$ S.E.M., $n = 98$) as compared to controls (57.6 $\pm$ 1 nm; mean $\pm$ S.E.M., $n = 140$), similar to what we observed in one-cell stage embryos (Fig. 4e). Altogether, our data strongly suggest that Ist1 plays a key role in ILV formation, which is most pronounced under conditions where flux through the system is elevated.

We also examined a potential role for Ist1 in endosomal recycling. The C. elegans cargo adaptor for intracellular Wnt trafficking (Mig14) is a well-characterized substrate for retromer-dependent membrane recycling in the germline[45]. In contrast to the potent effect of inhibiting retromer function on endosomal recycling, inhibition of Ist1 had no effect on Mig14 distribution or trafficking in the germline or early embryo (Supplementary Fig. 3c, d). Altogether, these data suggest a specific role for Ist1 in MVE biogenesis in C. elegans.

To define how Ist1 contributes to MVE formation, we first purified the protein from embryo extracts and identified binding partners using solution mass spectrometry. Consistent with prior work, Ist1 co-purified with Did2/CHMP1, but we also identified other ESCRT-III subunits, including Vps20 and Vps32 (Supplementary Table 1). Purification of Did2/CHMP1 defined an identical set of interaction partners, and immunoprecipitation of Vps32 revealed the presence of Ist1 and Did2/CHMP1 peptides, suggesting that both Ist1 and Did2 associate with factors that initiate polymerization of ESCRT-III spiral filaments (Supplementary Table 1). Each interaction was confirmed by immunoblot (Supplementary Fig. 4a). We next examined how Ist1 depletion affects the distribution of early-acting and late-acting ESCRT complexes using confocal microscopy. Similar to controls, we found that embryos depleted of Ist1 or the core ESCRT-III subunit Vps2 accumulated ESCRT-0 on Cav1 positive endosomes in early-stage zygotes (Fig. 5a, b and Supplementary Fig. 4b). In contrast, at an identical timepoint of embryo development, we failed to observe equivalent levels of the ESCRT-III subunit Vps32 on endosomes in the absence of Ist1 (Fig. 5c, d). Instead, Vps32 only weakly associated with Cav1 containing endosomes following Ist1 inhibition mediated by RNAi or in embryos expressing a truncated mutant that leads to Ist1 instability (Fig. 5e). Importantly, under these conditions, Vps32 stability was not affected, suggesting instead that Ist1 directly regulates ESCRT-III complex assembly (Supplementary Fig. 4c).

Consistent with prior work[5, 46, 47], inhibition of Vps2 leads to the excess accumulation of Vps32 on endosomal membranes, likely due to diminished Vps4 recruitment, which normally promotes filament disassembly (Fig. 5c, e). To determine whether Ist1 acts upstream of Vps2, we depleted Vps2 in embryos impaired for Ist1 function and examined Vps32 distribution early in zygotic development using STED imaging. Under these conditions, we found that Vps32 failed to accumulate on endosomes to the same degree as observed following Vps2 depletion in control embryos (Fig. 5e, f and Supplementary Fig. 4d). Instead, we observed Vps32 at small subdomains on these endosomes (Fig. 5f and Supplementary Fig. 4d). These data suggest that Ist1 acts upstream of Vps2 to facilitate or stabilize Vps32 filament assembly and support ILV formation during rapid MVE formation.

The reduced size of ILVs following Ist1 inhibition suggests it functions beyond simply recruiting Vps32. To gain further insight into the role of Ist1, we examined the distribution of early-acting

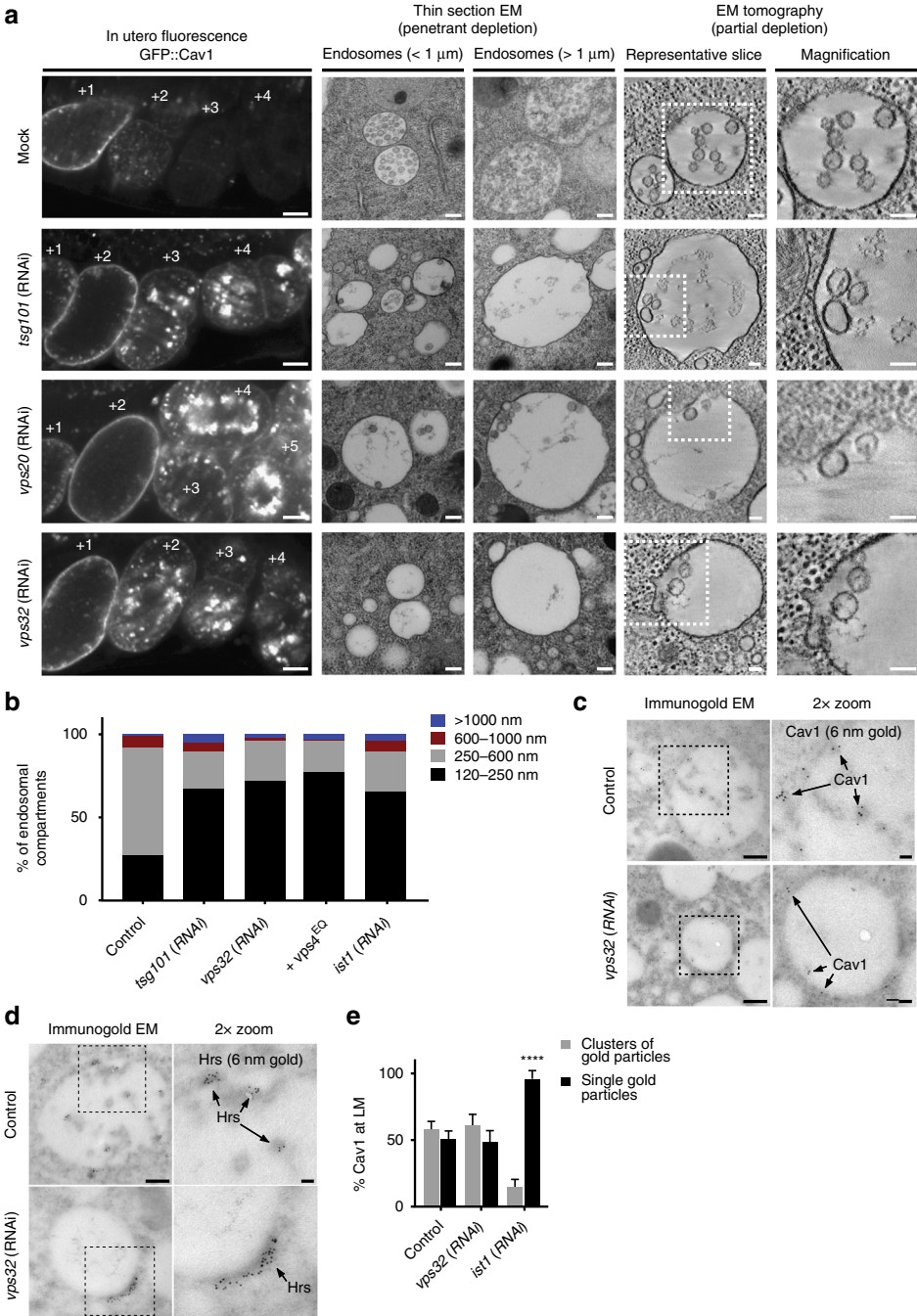

**Fig. 3** ESCRT-III is required for membrane deformation and cargo retention on MVEs in vivo. **a** Control ($n = 35$, mock-treated animals) and RNAi-treated animals ($n = 12$, Tsg101 depleted animals; $n = 11$, Vps20 depleted animals; $n = 35$, Vps32-depleted animals) expressing GFP::Cav1 were imaged using fluorescence (left panels) and electron microscopy in utero. The effects of penetrant ESCRT subunit depletion were analyzed by thin-section electron microscopy (middle panels; $n = 54$, mock-treated animals; $n = 77$, Tsg101 depleted animals; $n = 63$, Vps20 depleted animals; $n = 74$, Vps32-depleted animals), and partial depletions were conducted to enable ILV formation, which was examined by electron tomography (right panels; $n = 118$, mock-treated; $n = 15$, Tsg101 depletion; $n = 10$, Vps20 depletion; $n = 32$, Vps32 depletion). Representative images are shown. Bars, 10 μm (left panels, fluorescence imaging), 200 nm (middle panels, thin-section EM), and 100 nm (right panels, tomographic slices). **b** The size distribution of endosomal compartments in the one-cell stage embryo was determined for control animals ($n = 86$), ESCRT subunit depleted animals ($n = 212$, Tsg101 depleted animals; $n = 77$, Vps32-depleted animals; $n = 110$, Ist1-depleted animals), and transgenic animals expressing Vps4$^{E219Q}$ ($n = 112$). **c** The distribution of Cav1 at endosomes was determined by immunogold labeling in control ($n = 122$) and Vps32-depleted one-cell stage embryos ($n = 40$). Micrographs shown are representative of the endosomal compartments analyzed for each condition. Bars, 200 nm (left panels) and 50 nm (right, zoomed panels). **d** The distribution of Hrs (ESCRT-0) at endosomes was determined by immunogold labeling in control ($n = 42$) and Vps32-depleted one-cell stage embryos ($n = 36$). Micrographs shown are representative of the endosomal compartments analyzed for each condition. Bars, 200 nm (left panels) and 50 nm (right, zoomed panels). **e** Quantification of Cav1 distribution in control ($n = 37$) and ESCRT-III subunit depleted embryos ($n = 25$, Vps32 depletion; $n = 33$, Ist1 depletion) was determined using immunogold labeling, as described in panel 1 f. Standard error of the mean is shown in each case. ****$p < 0.0001$, using a two-way ANOVA with Dunnett's test

ESCRT components using immunogold electron microscopy following Ist1 inhibition. Strikingly, the distribution of ESCRT-0 was dramatically altered in the absence of Ist1. Instead of accumulating on subdomains at the endosome limiting membrane, more than 80% of gold labeling occurred on the limited number of ILVs that formed (Fig. 6a, b). Altogether, these data suggest that Ist1 regulates ESCRT-III assembly to both restrict deposition of early-acting ESCRT components into ILVs and sequester ubiquitin-modified cargoes within endosomal subdomains destined for internalization. Consistent with this idea, we found using STED imaging that ESCRT-0 co-localized with Vps32 endosomal subdomains that were able to form in the absence of Ist1, contrasting their juxtaposed distribution that was

most often observed under control conditions (Fig. 6c–e). Altogether, our findings support a model in which early-acting ESCRT complexes promote assembly of adjacent ESCRT-III complexes, which encircle cargoes and promote their specific internalization into ILVs.

## Discussion

Both early-acting and late-acting components of the ESCRT machinery have been demonstrated to promote membrane deformation on model membranes[14–17, 48]. These results have been difficult to reconcile, largely due to the absence of an appropriate model system to examine the problem in an in vivo setting. If ESCRT-I and ESCRT-II were sufficient to drive membrane invagination, depletion of ESCRT-III subunits should not prevent formation of nascent ILVs. Our analysis of the protein requirements for ILV formation contradicts this prediction, and instead supports a model in which ESCRT-III assembly directs membrane bending. However, the specific mechanisms that enable ESCRT-III to promote ILV formation remain unclear. Both theoretical and in vitro studies predict that molecular crowding of integral membrane cargoes could support bud formation[49]. Consistent with this idea, we demonstrate that ESCRT-III assembly is required to retain cargoes within membrane subdomains on endosomes. Moreover, we show evidence that this

**Table 2 Intralumenal vesicle (ILV) diameter statistics**

| | |
|---|---|
| Control | 57 ($\pm$1) nm, $n = 534$ |
| *tsg101* (RNAi) | 99 ($\pm$4) nm, $n = 43$ |
| *vps20* (RNAi) | 87 ($\pm$2) nm, $n = 80$ |
| *vps32* (RNAi) | 82 ($\pm$7) nm, $n = 18$ |
| *ist1* (RNAi) | 34 ($\pm$1) nm, $n = 131$ |

On the basis of electron tomography studies, the sizes of ILVs in control and partially depleted one-cell stage embryos were determined. S.E.M. is reported for each

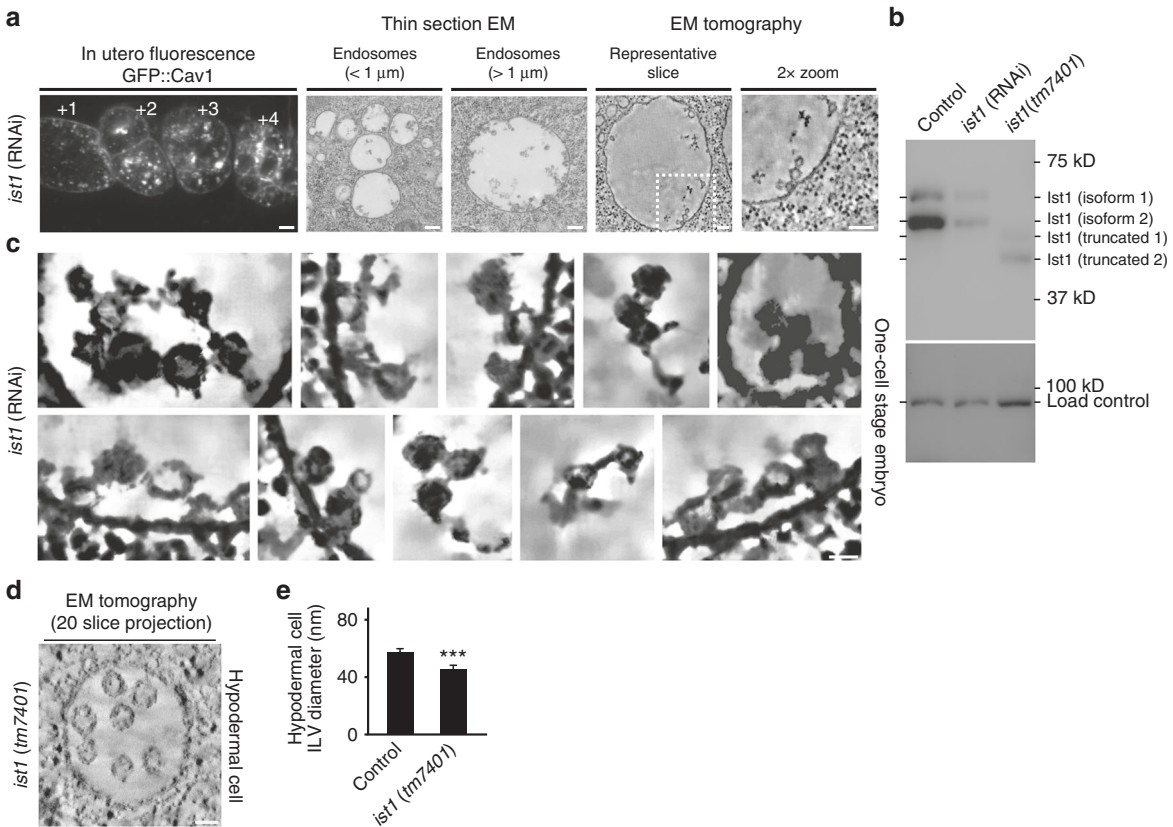

**Fig. 4** Ist1 regulates MVE biogenesis. **a** Animals expressing GFP::Cav1 were depleted of Ist1 using RNAi and imaged using fluorescence (left panels; $n = 20$ animals) and electron microscopy. Both thin-section electron microscopy (middle panels; 313 MVEs analyzed) and electron tomography (right panels; $n = 13$ MVEs analyzed) were conducted to visualize MVE and ILV morphology. Representative images are shown. Bars, 10 μm (left panels, fluorescence imaging), 200 nm (middle panels, thin-section EM), and 100 nm (right panels, tomographic slices). **b** Extracts were generated from control, Ist1-depleted and *ist1* (*tm7401*) mutant animals and separated by SDS–PAGE, followed by immunoblot analysis using antibodies directed against Ist1. Two splice variants have been identified for Ist1, each of which was detected. **c** Regions extracted from several tomograms described in panel **a** were processed for noise reduction and feature enhancement. Bar, 25 nm. **d** Representative MVE found in a hypodermal cell ($n = 12$) from an animal harboring the *ist1* (*tm7401*) mutation imaged using electron tomography. Bar, 50 nm. **e** Quantification of ILV diameter found in control and *ist1* (*tm7401*) hypodermal cells. Standard error of the mean is reported in each case. ***$p < 0.001$, using a *t*-test

action may occur independently of ESCRT-0 subdomain formation, which fail to correspond to the precise sites of membrane budding or ESCRT-III assembly in vivo[50]. One explanation for these findings relates to the low affinity of the early-acting ESCRT complexes for ubiquitin-modified cargoes[51, 52]. Instead of acting to directly concentrate cargoes for internalization into ILVs, the early-acting ESCRT complexes may simply limit their diffusion transiently[53], enabling capture within adjacent ESCRT-III spiral

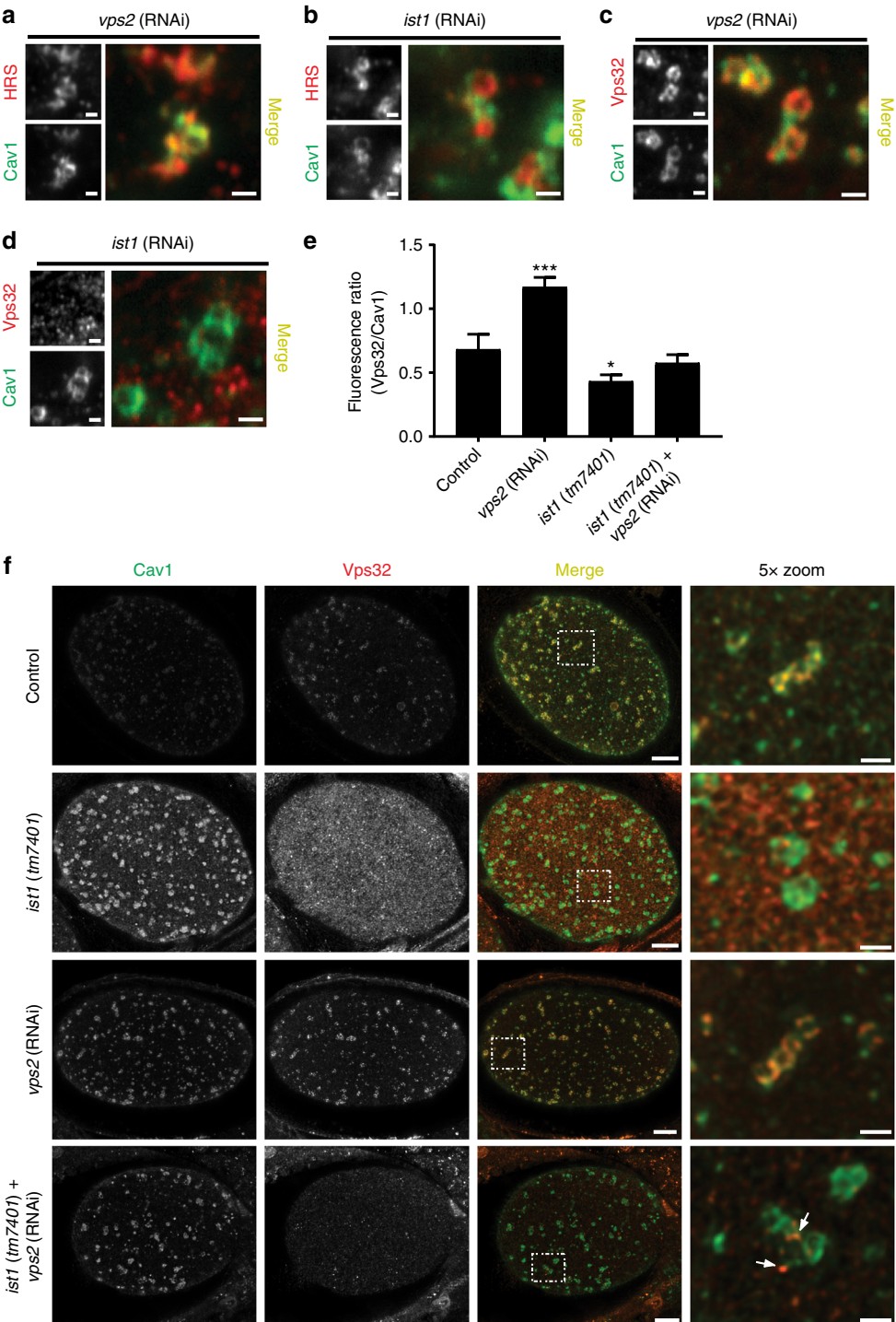

**Fig. 5** Ist1 controls Vps32 assembly on endosomes. **a–d** One-cell stage embryos expressing GFP::Cav1 and depleted of either Vps2 ($n = 13$, each condition) or Ist1 ($n = 11$, each condition) were fixed and stained using antibodies directed against GFP and either Hrs or Vps32 and imaged using confocal microscopy. Representative images are shown. Bars, 1 μm. **e** Quantification of the relative fluorescence intensities of Vps32 and Cav1 under various conditions based on STED imaging (see panel **f**). Standard error of the mean is reported in each case. ***$p < 0.001$ as compared to control, using a $t$-test; *$p < 0.05$ as compared to control, using a one-way ANOVA with Dunnett's test. **f** One-cell stage control ($n = 10$, each condition) and ist1 (tm7401) mutant embryos ($n = 20$, each condition) expressing GFP::Cav1 in the presence or absence of Vps2 were fixed and stained using antibodies directed against GFP and Vps32 and imaged using STED microscopy. Arrows highlight the presence of subdomains containing Vps32. Representative images are shown. Bars, 5 μm (left panels) and 1 μm (right, zoomed panels)

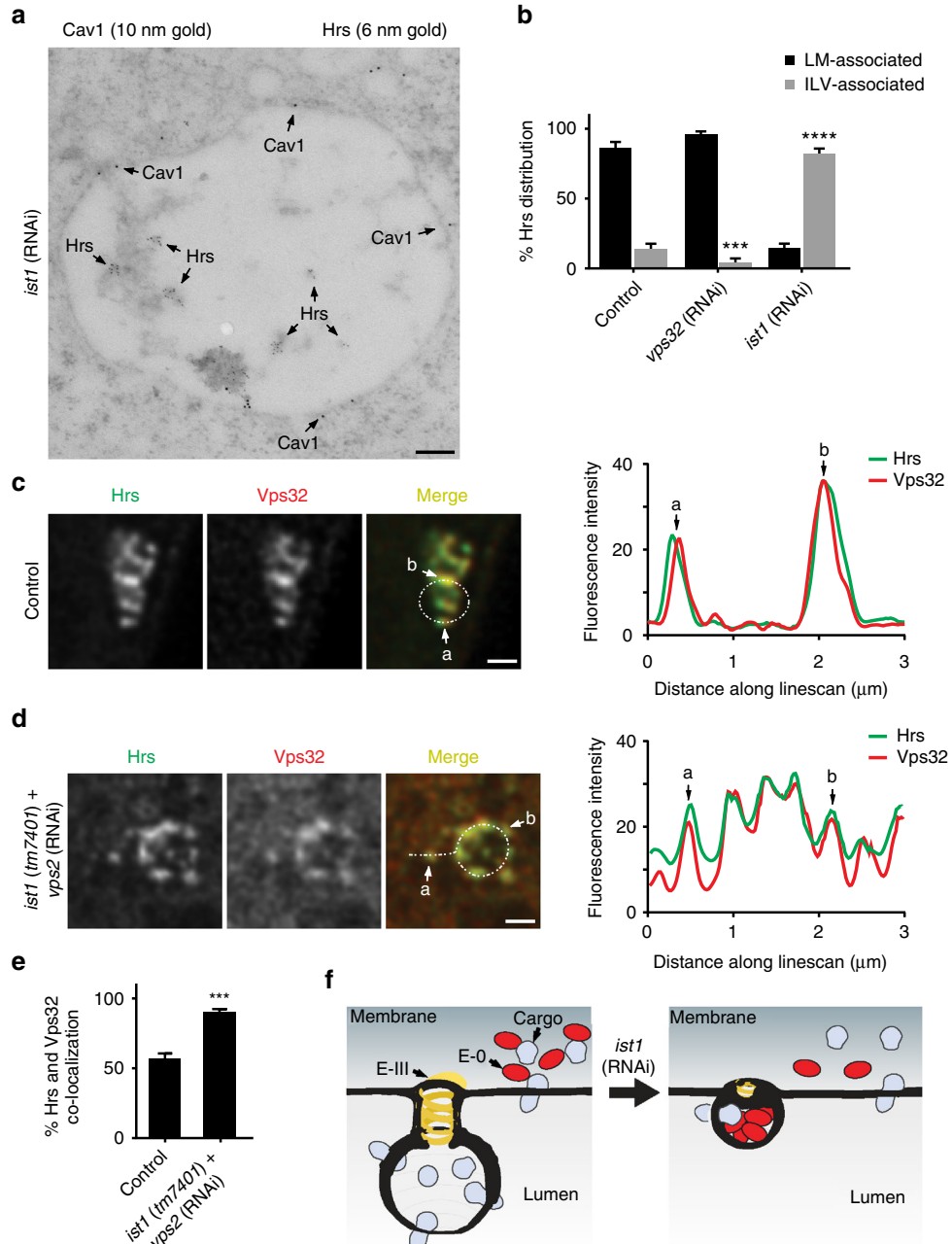

**Fig. 6** Ist1 inhibits the incorporation of ESCRT-0 into ILVs. **a** Immunogold labeling was used to determine the distributions of Cav1 (10 nm gold) and Hrs (6 nm gold) in one-cell stage embryos depleted of Ist1 following high-pressure freezing. The image shown is representative of 64 MVEs analyzed. Bar, 100 nm. **b** Quantification of the distribution of Hrs at the limiting membrane and within MVEs in control ($n = 31$), Vps32-depleted ($n = 36$), and Ist1-depleted ($n = 27$) one-cell stage embryos. Standard error of the mean is reported in each case. ***$p < 0.0005$; ****$p < 0.0001$ compared to control using a two-way ANOVA with Dunnett's test. **c, d** Control ($n = 104$) and *ist1 (tm7401)* mutant embryos ($n = 111$) expressing GFP::Cav1 and depleted of Vps2 using RNAi were fixed and stained using antibodies directed against Hrs and Vps32. Samples were imaged using STED microscopy, and representative images are shown. Linescan analysis was conducted to measure the relative distributions of Hrs and Vps32 at endosomes ($n = 76$ linescans per condition). Specific sites along each linescan are highlighted. Bars, 500 nm. **e** Quantification of co-localized Hrs and Vps32 under conditions described in panel **d**. Standard error of the mean is reported in each case. ***$p < 0.001$ using a *t*-test. **f** A model of MVE biogenesis in the presence and absence of Ist1. Under normal conditions, cargoes are sorted into membrane subdomains that are internalized via ESCRT-III activity. However, following depletion of Ist1, cargo retention within ESCRT-III subdomains is impaired and ESCRT-0 becomes engulfed into ILVs of reduced size

assemblies that are nucleated by the upstream machinery. Such a model obviates a necessity for direct transfer between early-acting and late-acting ESCRT complexes.

However, the efficiency of a passive transfer system is questionable in the context of individual ILV budding events, as cargoes would likely escape too rapidly[54, 55]. Instead, our data

suggest that ILVs bud repetitively from individual membrane subdomains, which would facilitate cargo internalization. Furthermore, continuous ILV budding depends on cycles of Vps4-mediated ESCRT-III disassembly. Previous work demonstrated that Vps4 does not regulate the disassembly of early-acting ESCRT components[40], suggesting that ESCRT subdomains are

stable and capable of continuously nucleating ESCRT-III polymers for numerous rounds of ILV biogenesis. These data raise the question of how other cargoes, including those that recycle, elude deposition into ILVs. On the basis of a recent study, other endosomal subdomains may directly compete with ESCRT complex assembly to enable discrimination between degradative cargo sorting and cargo recycling[56]. In the absence of specific retromer subunits, ESCRT coverage on endosomes expands dramatically, leading to the degradation of cargoes that normally recycle.

Our data also indicate that proper ESCRT-III assembly enables cargo exclusion, preventing consumption of upstream ESCRT complexes within ILVs during rapid MVE biogenesis. We further demonstrate a role of Ist1 in regulating this activity of the ESCRT-III complex. The mechanism by which Ist1 functions in this context remains unclear, but is likely linked to its interactions with Vps20 and Vps32. In the absence of Ist1, Vps32 polymer assembly on endosomes is reduced, inhibiting ILV formation and reducing ILV size. These data contrast findings in yeast and mammalian cells, where inhibition of Ist1 was shown to have no impact on Vps32 distribution or degradative cargo sorting[23, 24, 26]. However, in those contexts, cells were not stimulated to synthesize new MVEs. Our findings directly demonstrate that Ist1 plays a critical role during rapid MVE formation, potentially through its co-assembly with nucleators of ESCRT-III filament formation. In doing so, Ist1 facilitates Vps32 polymerization and simultaneously inhibits diffusion of ESCRT-0 into areas targeted by ESCRT-III spiral filaments for internalization. In parallel, Ist1 has also been suggested to inhibit Vps4 assembly[24, 40], and removal of Ist1 may dysregulate Vps4-mediated depolymerization of ESCRT-III filaments, thereby altering the timing of ILV scission and reducing ILV size.

## Methods

**Antibodies, immunofluorescence, and live animal imaging**. Polyclonal antibodies directed against *C. elegans* STAM, Hrs, Tsg101, Vps20, Vps32, Vps2, Did2/CHMP1, and Ist1 have been described and were used at a final concentration of 1 µg/mL[20, 29, 31, 39, 51], and antibodies against GFP (used at 1 µg/mL) were obtained from commercial sources (TP401, Torrey Pines Biolabs, Secaucus, NJ; 600-101-215, Rockland, Limerick, PA).

Confocal images were acquired on a swept-field confocal microscope (Nikon Ti-E) equipped with a Nikon 60×, 1.4NA Planapo oil objective lens and a CoolSNAP HQ2 CCD camera. Immunofluorescence of fixed embryos was performed using directly labeled rabbit antibodies at a final concentration of 1 µg/mL[31]. Acquisition parameters and image analysis was performed using Nikon Elements software. For live imaging, transgenic animals were immobilized within a 4 µl suspension of polystyrene beads (Polysciences, 2.5% by volume, 0.1 µm diameter) that was placed onto the center of a 10% agarose pad and gently compressed with a coverslip[32]. All images shown are representative of experiments conducted more than three times independently.

**C. elegans growth, maintenance and RNA interference**. *C. elegans* strains used were derived from the Bristol strain N2. The deletion strain, *ist-1(tm7401)*, was provided by the Mitani Lab through the National Bio-Resource Project (Japan). Strains stably expressing Rme2::GFP, GFP::Cav1, Mig14::GFP, mCherry:His58, and GFP::Tsg101 were generated using biolistic transformation and have been described previously[28, 37, 57, 58]. Animals expressing mCherry:STAM were generated using MosSCI[59], and animals expressing Vps4$^{E219Q}$ were generated by biolistic transformation[60]. Double-stranded RNA (dsRNA) was synthesized from PCR templates prepared using primers to amplify N2 genomic DNA or N2 cDNA. For RNAi experiments, early L4 stage hermaphrodites were soaked in dsRNA for 24 h at 20 °C within a humidified chamber and then transferred onto bacteria expressing the same dsRNA for an additional 48 h prior to analysis. To ensure the animals were still generating embryos several days after beginning RNAi treatment, we incubated them with wild type *C. elegans* males. The addition of males had no effect on knockdown efficiency in early embryos, which are transcriptionally silent until the ~4 cell stage[61].

**High-pressure freezing, freeze substitution, and embedding**. *C. elegans* samples were loaded into 1-hexadecane-coated, 100 µm deep aluminum sample holders (Technotrade, Manchester, NH) filled with a suspension of OP50 bacteria, and frozen using a Balzers HPM 010 high-pressure freezer. Samples were substituted in

~3 h, in either 1% OsO$_4$, 1% H$_2$0 in acetone for ultrastructural studies, or 0.2% uranyl acetate in acetone[62] for immunogold labeling. Resin infiltration was accomplished by centrifugation of the animals through increasing concentrations of either LR White hard grade or epoxy EMbed 812[62], and polymerization was carried out in an oven at 50 °C (LR white) or 60 °C (epon) for 24 h.

Animals were oriented on beam capsules to facilitate sectioning along the longitudinal axis, and thin (80 nm) sections were collected on Pioloform film copper slot grids. The epon-embedded samples were post-stained with 8% uranyl acetate in 50% ethanol for 10 min at room temperature, followed by lead citrate for 10 min at room temperature. For tomography samples, semi-thick (300 nm) sections were collected on grids, post-stained and incubated in drops of 10 nm colloidal gold particles (Electron Microscopy Sciences, Hatfield, PA) that serve as fiducial markers for aligning tomograms.

**Immunogold labeling**. LR-White-embedded samples were thin-sectioned (80 nm) onto carbon-coated Pioloform film grids. Post-embedding immunogold labeling was performed using the Aurion method (http://www.aurion.nl/the_aurion_method/Post_embedding_conv.php). Following immunogold labeling, the samples were stained with 4% uranyl acetate in H$_2$O for 2 min at RT. The distribution of gold particles was assessed by measuring distances between individual particles and categorizing them as being individual (no other gold particles within 20 nm) or as a cluster/subdomain (two or more gold particles within 20 nm of one another).

**Electron microscopy, tomography, and reconstructions**. Electron micrographs of 80 nm sections were collected on a Phillips CM120 80 kV TEM equipped with an AMT Biosprint 12 series digital camera or a MegaView III digital camera (Olympus Soft Imaging Solutions, Münster, Germany). Single-axis and dual-axis tilt series were collected using SerialEM software[63] on a 300 kV Tecnai TF-30 equipped with a Gatan 2k × 2k Ultrascan camera. Tilt series images were captured at magnifications from 12kX (0.94 nm/pixel) to 23kX (0.467 nm/pixel) at 1° increments from −65° to + 65°. Reconstruction and modeling of tilt series was carried out using IMOD software[64, 65]. Tomograms were reconstructed using back projection and/or simultaneous iterative reconstruction technique (SIRT) using 5–10 iterations of standard settings in IMOD version 4.7.13. Modeling of membranes was performed in 3dmod by drawing contours between the leaflets of the lipid bilayer.

To characterize tethers between ILVs, each electron tomography tilt series was aligned using the IMOD package[64] with a combination of fiducial-based and patch-based approaches. Three-dimensional densities were generated using SIRT as implemented in Tomo3D[66]. This reconstruction technique tends to show a better definition of cell membranes and fine connections as compared to alternative Fourier or weighted back-projection methods. Non-local means filtering[67] was applied to reduce noise. Histogram equalization was performed to further enhance the visibility of thin connections between the ILVs.

**Immunoprecipitation and mass spectrometry**. Adult hermaphrodites were grown in liquid culture, and embryonic lysates were generated for immunoprecipitation studies[29]. For mass spectrometry, proteins were reduced with 5 mM Tris (2-carboxyethyl)phosphine hydrochloride (Sigma-Aldrich, St. Louis, MO) and alkylated. Proteins were digested for 18 h at 37 °C in 2 M urea, 100 mM Tris pH 8.5, 1 mM CaCl$_2$ with 2 µg trypsin (Promega, Madison, WI). Analysis was performed on a Thermo LTQ Orbitrap or Thermo LTQ Orbitrap XL using an in-house built electrospray stage[68]. Protein and peptide identification and protein quantitation were done with the Integrated Proteomics Pipeline - IP2 (Integrated Proteomics Applications, Inc., San Diego, CA). Tandem mass spectra were extracted from raw files using RawExtract 1.9.9.2 and were searched against Wormbase protein database WS257 with reversed sequences using ProLuCID[69]. The search space included fully tryptic and half-tryptic peptide candidates with a fixed modification of 57.02146 on cysteine. Peptide candidates were filtered using DTASelect with a protein false discovery of 1%[70].

**Statistical considerations**. In all cases, sample sizes were chosen on the basis of the smallest effect that could be reasonably measured. No individual samples were excluded from analysis. Statistical tests used include *t*-tests and ANOVA, depending on the number of samples being compared.

**Data availability**. The data that support the findings of this study are available from the corresponding author upon reasonable request.

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

## Acknowledgements

This work was supported in part by grants from the NIH (GM088151 to A.A., R01GM115972 to N.V., and 8 P41 GM103533 to J.R.Y.). Additional support for this research was provided by the Office of the Vice Chancellor for Research and Graduate Education at the University of Wisconsin-Madison with funding from the Wisconsin Alumni Research Foundation. E.B.F. was supported by NIH National Research Service Award T32 GM07215. We thank members of the Audhya lab for suggestions and critically reading this manuscript, Marisa Otegui for useful discussions, Amber Schuh for generating transgenic animals, and Ben August for technical assistance.

## Author contributions

Conceived and designed experiments: E.B.F., A.A. Performed experiments and analyzed data: E.B.F., R.S., J.J.M., N.V. and A.A. Contributed reagents/materials/analysis tools: N. V., J.R.Y. and A.A. Wrote the paper: E.B.F. and A.A.

## Additional information

**Competing interests:** The authors declare no competing financial interests.

