## [Peer Review File · Nature Communications]

Reviewers' Comments:

Reviewer #1:

Remarks to the Author:

The pieces of STED imaging data reported in Figures 5 and 6 of this manuscript and their interpretation appear reasonable, but straightforward questions remain, which should be answered by the authors in their revision:

In Fig. 5e, it is expected that the higher resolution of STED images allows more quantitative ratio determinations, but it is stated that confocal data is shown in all of Fig. 5 a-d. The authors should confirm this is the case, and comment if necessary. Figure 5f indeed shows punctate Vps32 domains of slightly sub-diffraction extent.

The linescans in Fig. 6c and d and the derived degrees of co-localization (Fig. 6e) make use of the ability to detect spatial shifts by sub-diffraction dimensions with enhanced fidelity. The number of linescans to yield the quantification in panel e should be stated. A complete assessment of the validity of these results can only be made, as is often the case for such experiments, with full access to raw data.

Reviewer #2:

Remarks to the Author:

The manuscript by Frankel et al describes the development of a system to study de novo ILV formation in MVEs using *C. elegans* early embryogenesis, and reports on discoveries related to ESCRT-III and IST1 function. One remarkable aspect of the system described is that the authors find no MVEs in *C. elegans* oocytes, but find numerous MVEs in zygotes shortly after fertilization. These findings provide a powerful system to study the formation of ILVs because their genesis is controlled by a developmental switch, and their progress can be followed on a timescale of minutes after their formation (or attempted formation). Primarily using a combination of RNAi-mediated depletions, transmission electron microscopy, electron tomography, and immunoEM, the authors show that newly formed ILVs appear connected by threads and appear clustered toward one side of an MVE, suggesting that ILVs form in bursts from a defined region of the MVE. The authors then move on to characterize a ubiquitylated cargo that is normally degraded in the early embryo (CAV-1), following its incorporation into peripheral clusters and inclusion in ILVs by immunoEM. They propose that ESCRT-III is required to cluster CAV-1 into subdomains that would normally lead to their inclusion in ILVs, with no apparent effect on ESCRT-0 clustering. The most interesting data in the paper focuses on a unique ILV-related role for IST1, finding that loss of IST1 via RNAi or mutation reduced ILV production and led to abnormally small residual ILVs, while residual ILVs after depletion of other ESCRT components leads to enlarged ILVs. Loss of IST1 also strongly impaired recruitment of VPS32 to the endosomes, and led to inappropriate inclusion of ESCRT-0 into ILVs. The authors propose that IST1 is a key regulator of ESCRT-III assembly during ILV formation. Overall I found the data to be of very high quality and I expect the unique findings in this system to be of wide interest to a cell biological audience.

Issues to be addressed:

(1) Many interpretations are based upon the clustering of gold particles in immunoEM. It was not entirely clear to me how the quantifications were performed, and how many clustered particles are needed to declare a "subdomain". Are two particles in close proximity a subdomain? Several key data sets of this type lack any statistical analysis (e.g. figs 3d, 5b)

(2) The statement on the first page of results that GFP::CAV-1 “fluorescence intensity in oocytes remained stable over several hours, suggesting an absence of ESCRT-mediated protein turnover during this stage of development.” is not valid. In oocytes GFP::CAV-1 is in secretory cortical granules awaiting secretion after fertilization, and is not in endosomes, so GFP::CAV-1 fluorescence intensity has no bearing on the question of whether integral membrane cargo can be degraded in oocytes (e.g. Kimura 2012 PMID: 22992455, Olson 2012 PMID: 22908315, Bembenek 2007 PMID: 17913784).

However, other data that is presented, such as the absence of visible ILVs by EM, and the failure to find endosomes bearing multiple ESCRT components by IF, does support the overall point that embryos undergo de novo MVE generation. I recommend removing the GFP::CAV-1 related sentence.

(3) I don't think Fig 1F measured accumulations at subdomains as stated here: “At later stages of development in the one-cell embryo, GFP::CAV-1 at the endosomal limiting membrane exhibited a more restricted distribution, accumulating on distinct subdomains (Fig. 1f).” I did not find any data on this point in the paper.

(4) This section header seems overstated: “Repetitive formation of ILVs occurs at individual endosomal microdomains during rapid MVE biogenesis”. The main text is more appropriate on this point and does not make such a strong interpretation, instead saying the data is “consistent with” this repetitive formation of ILVs occurs at individual endosomal microdomains.

(5) I don't understand how finding that ESCRT-III is required to form ILVs: “Importantly, depletion of Vps32 potentially blocked membrane bending on endosomes, indicating that ESCRT-III activity is required to initiate ILV formation in vivo (Fig. 3a, c, e).” means that ESCRT-I and ESCRT-II play no role in making ILVs: “These data contrast prior in vitro studies, which argued that ESCRT-I and ESCRT-II play an active role in this process.” To show this the authors would need to deplete ESCRT-I and ESCRT-II and show normal ILV formation still occurs. Rather it appeared that *tsg101* depletion did affect ILV number and size in Fig 3.

(6) The paper is not very consistent on nomenclature, using approved *C. elegans* protein names (all CAPS with a dash) in some places, but not others (especially figures). Not sure what journal rules dictate, but protein and gene names should at least be consistent throughout the paper.

Reviewer #3:

Remarks to the Author:

Ist1 regulates ESCRT-III assembly and function during multivesicular endosome biogenesis

The authors take advantage of a developmental transition during *C. elegans* development, the oocyte-to-embryo transition, to probe the roles of ESCRTs therein. This system appears to offer multiple strengths, including synchronized waves of cargo sorting and multivesicular endosome biogenesis. Using this system the authors make several observations that are in striking contrast to previous results. One conclusion is that ILVs form with a tethered morphology, suggesting that they bud continuously from a microdomain. Another observation is that perturbation of IST1 manifests in a defect in the biogenesis of ILVs, apparently the result of defective recruitment of ESCRT subunits. This is an intriguing system with which to address the impact of endosomal sorting on development and it has highlighted some contentious aspects within the field. Unfortunately, it fails to reveal whether the findings are specific aspects of endosomal sorting unique to this particular developmental transition. This results in significant reduction in the enthusiasm of this reviewer.

1) Multiple groups have observed no defect in MVB sorting upon perturbation of IST1 alone, however synthetic genetic interactions have been observed in which IST1 perturbation contributes to an MVB sorting defect and previous studies have revealed Ist1 can have both a positive or negative impacts on MVB sorting. Thus, there is a significant amount of data supporting the conclusion that Ist1 plays a role in ILV biogenesis. While implicating Ist1 in ESCRT-III function is not novel, the current studies make unique observations in this particular biological context suggesting a larger contribution than previously appreciated. However, reconciliation of the current and previous observations is lacking and should be addressed. Does the unique environment of the germline (e.g. altered expression of other ESCRT factors) contribute to this discrepancy? Is the previously ascribed role for Ist1 in recycling an artifact? Why have the assertions that Ist1 directly regulates ESCRT-III assembly and/or function been incorrect prior to the present work? The current language appears to be imparting undue significance to the present findings without acknowledging any caveats about how broadly the findings may be interpreted.

2) The current observations support the conclusion that ESCRT-I and -II do not play a role in membrane deformation. While this conclusion is not surprising to those that have followed the field, it is unclear whether unique aspects of this experimental system impact the generality of this conclusion.

3) The observation that multiple ILVs have contact with one another may suggest that multiple vesicle formation events are linked (without intermittent scission). Whether this is something that occurs during states of high flux or represents a universal feature of ILV biogenesis that has never been observed before is not clear. These studies are suggestive but do not directly address whether stable microdomains of ESCRT function exist.

4) Demonstration that Ist1 can function upstream of Vps2 in sorting events during oocyte-to-embryo transition needs to compare Vps32 localization in contexts where Vps4 function is perturbed. A possible explanation for failure of Vps32 to accumulate to the same degree in *ist1* vs *vps2* is that *vps2* is defective for ESCRT-III disassembly, while Ist1 may not be (based on previous data). If IST1 is epistatic to VPS2 then *ist1* and *ist1 vps2* morphologies would be indistinguishable.

Manuscript # NCOMMS-17-07153-T; Revision submission date: July 18, 2017

Title: "Ist1 regulates ESCRT-III assembly and function during multivesicular endosome biogenesis"

Response to Comments made by the Reviewers:

Reviewer #1: The pieces of STED imaging data reported in Figures 5 and 6 of this manuscript and their interpretation appear reasonable, but straightforward questions remain, which should be answered by the authors in their revision: In Fig. 5e, it is expected that the higher resolution of STED images allows more quantitative ratio determinations, but it is stated that confocal data is shown in all of Fig. 5 a-d. The authors should confirm this is the case, and comment if necessary. Figure 5f indeed shows punctate Vps32 domains of slightly sub-diffraction extent.

We confirm that the data shown in Fig. 5 a-d were acquired using confocal microscopy. However, quantifications of fluorescence ratios shown in Fig. 5e are based on STED microscopy, as stated in the figure legend. STED images are shown in Fig. 5f.

Reviewer #1: The linescans in Fig. 6c and d and the derived degrees of co-localization (Fig. 6e) make use of the ability to detect spatial shifts by sub-diffraction dimensions with enhanced fidelity. The number of linescans to yield the quantification in panel e should be stated. A complete assessment of the validity of these results can only be made, as is often the case for such experiments, with full access to raw data.

We apologize for this oversight. We have now stated the number of linescans analyzed in the figure legend for Fig. 6 (more than 75 linescans for each condition). We can include an excel spreadsheet with the raw data from these linescans at the discretion of the journal (however, to our knowledge, such data are typically not included in supplementary material).

Reviewer #2: The manuscript by Frankel et al describes the development of a system to study de novo ILV formation in MVEs using C. elegans early embryogenesis, and reports on discoveries related to ESCRT-III and IST1 function. One remarkable aspect of the system described is that the authors find no MVEs in C. elegans oocytes, but find numerous MVEs in zygotes shortly after fertilization. These findings provide a powerful system to study the formation of ILVs because their genesis is controlled by a developmental switch, and their progress can be followed on a timescale of minutes after their formation (or attempted formation). Primarily using a combination of RNAi-mediated depletions, transmission electron microscopy, electron tomography, and immunoEM, the authors show that newly formed ILVs appear connected by threads and appear clustered toward one side of an MVE, suggesting that ILVs form in bursts from a defined region of the MVE. The authors then move on to characterize a ubiquitylated cargo that is normally degraded in the early embryo (CAV-1), following its incorporation into peripheral clusters and inclusion in ILVs by immunoEM. They propose that ESCRT-III is required to cluster CAV-1 into subdomains that would normally lead to their inclusion in ILVs, with no apparent effect on ESCRT-0 clustering. The most interesting data in the paper focuses on a unique ILV-related role for IST1, finding that loss of IST-1 via RNAi or mutation reduced ILV production and led to abnormally small residual ILVs, while residual ILVs after depletion of

other ESCRT components leads to enlarged ILVs. Loss of IST1 also strongly impaired recruitment of VPS32 to the endosomes, and lead to inappropriate inclusion of ESCRT-0 into ILVs. The authors propose that IST1 is a key regulator of ESCRT-III assembly during ILV formation. Overall I found the data to be of very high quality and I expect the unique findings in this system to be of wide interest to a cell biological audience. Overall I found the data to be of very high quality and I expect the unique findings in this system to be of wide interest to a cell biological audience.

We appreciate that the reviewer finds our study to be of very high quality and of wide interest to the field.

Reviewer #2: Many interpretations are based upon the clustering of gold particles in immunoEM. It was not entirely clear to me how the quantifications were performed, and how may clustered particles are needed to declare a “subdomain”. Are two particles in close proximity a subdomain? Several key data sets of this type lack any statistical analysis (e.g. figs 3d, 5b).

We apologize for the lack of clarity. As now stated in the text (page 6) and the legend for Fig. 1, two gold particles within 20 nm is defined as a subdomain. We have also elaborated on this issue in the methods section (pages 17-18). Additionally, we have added statistical analysis to all data sets reported in the manuscript (Figs., 1, 3, and 6), and we apologize for this oversight.

*Reviewer #2: The statement on the first page of results that GFP::*CAV-1* “fluorescence intensity in oocytes remained stable over several hours, suggesting an absence of ESCRT-mediated protein turnover during this stage of development.” is not valid. In oocytes GFP::*CAV-1* is in secretory cortical granules awaiting secretion after fertilization, and is not in endosomes, so GFP::*CAV-1* fluorescence intensity has no bearing on the question of whether integral membrane cargo can be degraded in oocytes (e.g. Kimura 2012 PMID: 22992455, Olson 2012 PMID: 22908315, Bembenek 2007 PMID: 17913784). However, other data that is presented, such as the absence of visible ILVs by EM, and the failure to find endosomes bearing multiple ESCRT components by IF, does support the overall point that embryos undergo de novo MVE generation. I recommend removing the GFP::*CAV-1* related sentence.*

The reviewer is entirely correct. We have revised the description of GFP::*CAV-1* trafficking, citing the studies highlighted by the reviewer (please see page 5).

*Reviewer #2: I don't think Fig 1F measured accumulations at subdomains as stated here: “At later stages of development in the one-cell embryo, GFP::*CAV-1* at the endosomal limiting membrane exhibited a more restricted distribution, accumulating on distinct subdomains (Fig. 1f).” I did not find any data on this point in the paper.*

Again, the reviewer is correct. We have revised Fig. 1f and the accompanying text in the manuscript (please see page 6).

Reviewer #2: This section header seems overstated: “Repetitive formation of ILVs occurs at individual endosomal microdomains during rapid MVE biogenesis”. The main text is more appropriate on this point and does not make such a strong interpretation, instead saying the data is “consistent with” this repetitive formation of ILVs occurs at individual endosomal microdomains.

We agree with the reviewer and the section header has been toned down (please see page 7).

Reviewer #2: I don't understand how finding that ESCRT-III is required to form ILVs: “Importantly, depletion of Vps32 potentially blocked membrane bending on endosomes, indicating that ESCRT-III activity is required to initiate ILV formation in vivo (Fig. 3a, c, e).” means that ESCRT-I and ESCRT-II play no role in making ILVs: “These data contrast prior in vitro studies, which argued that ESCRT-I and ESCRT-II play an active role in this process.” To show this the authors would need to deplete ESCRT-I and ESCRT-II and show normal ILV formation still occurs. Rather it appeared that tsg101 depletion did affect ILV number and size in Fig 3.

We apologize for the lack of clarity in this section. We have revised the text to indicate that upstream ESCRT complexes are necessary for the recruitment of ESCRT-III to multivesicular endosomes (please see pages 9-10). Nonetheless, our findings indicate that in the absence of ESCRT-III, formation of nascent ILVs fails, suggesting that ESCRT-III (and not ESCRT-I/II) promote membrane deformation necessary for inward vesicle budding.

Reviewer #2: The paper is not very consistent on nomenclature, using approved C. elegans protein names (all CAPS with a dash) in some places, but not others (especially figures). Not sure what journal rules dictate, but protein and gene names should at least be consistent throughout the paper.

We have gone through the entire manuscript and the set of figures to ensure consistency. We chose to use a nomenclature that is widely accessible to a broad readership.

Reviewer #3: The authors take advantage of a developmental transition during C. elegans development, the oocyte-to-embryo transition, to probe the roles of ESCRTs therein. This system appears to offer multiple strengths, including synchronized waves of cargo sorting and multivesicular endosome biogenesis. Using this system the authors make several observations that are in striking contrast to previous results. One conclusion is that ILVs form with a tethered morphology, suggesting that they bud continuously from a microdomain. Another observation is that perturbation of IST1 manifests in a defect in the biogenesis of ILVs, apparently the result of defective recruitment of ESCRT subunits. This is an intriguing system with which to address the impact of endosomal sorting on development and it has highlighted some contentious aspects within the field. Unfortunately, it fails to reveal whether the findings are specific aspects of endosomal sorting unique to this particular developmental transition. This results in significant reduction in the enthusiasm of this reviewer.

As detailed below, we now include several new pieces of data (Figs. 2g, 2h, 4d, and 4e) to indicate that the findings made by examining the oocyte-to-embryo transition are consistent with findings made in another tissue (the hypodermis). These new data strongly suggest that our findings highlight general features of ILV biogenesis.

Reviewer #3: Multiple groups have observed no defect in MVB sorting upon perturbation of IST1 alone, however synthetic genetic interactions have been observed in which IST1 perturbation contributes to an MVB sorting defect and previous studies have revealed Ist1 can have both a positive or negative impacts on MVB sorting. Thus, there is a significant amount of data supporting the conclusion that Ist1 plays a role in ILV biogenesis. While implicating Ist1 in ESCRT-III function is not novel, the current studies make unique observations in this particular biological context suggesting a larger contribution than previously appreciated. However, reconciliation of the current and previous observations is lacking and should be addressed.

We apologize for this oversight in our prior submission. We have now addressed this issue extensively, both in the results and discussion sections (in particular, please see pages 4, 11-12, and 15).

Reviewer #3: Does the unique environment of the germline (e.g. altered expression of other ESCRT factors) contribute to this discrepancy? Is the previously ascribed role for Ist1 in recycling an artifact? Why have the assertions that Ist1 directly regulates ESCRT-III assembly and/or function been incorrect prior to the present work? The current language appears to be imparting undue significance to the present findings without acknowledging any caveats about how broadly the findings may be interpreted.

This is an important point, and we are pleased that the reviewer raised this concern. To address some these comments, we conducted extensive studies to visualize MVE morphology by electron tomography in the hypodermis (see new figure panels 2g, 2h, 4d, and 4e), a tissue shown previously to possess relatively few MVEs (e.g., reduced flux through the ESCRT-dependent MVE biogenesis pathway as compared to the oocyte-to-embryo transition), similar to other model systems described in the literature. Based on our findings (e.g., ILVs continue to exhibit tethers between them and inhibition of Ist1 function leads to the formation of ILVs with significantly reduced diameter as compared to controls), we are confident that Ist1 plays an important and general role in ILV formation, which has previously been overlooked. Importantly, the high flux through the ESCRT-dependent MVE biogenesis pathway during the oocyte-to-embryo transition enabled us to identify these features for the first time and led to our detailed examination of MVE formation (one of the reasons we believe that the unique environment of the *C. elegans* germline is ideal to study native ESCRT function). It is noteworthy that our new data show that inhibition of Ist1 in the hypodermis fails to lead to an accumulation of ILVs near the limiting membrane (consistent with previous work). Nonetheless, examination of ILV size under this condition revealed a significant decrease in the diameter of ILVs, consistent with our findings during the oocyte-to-embryo transition (please see revised Fig. 4). The accumulation of ILVs at the endosome limiting membrane in one-cell stage embryos lacking Ist1 may be a result of a particularly high rate of MVE formation that is necessary during this stage of development, which we now

acknowledge in the text (please see pages 11-12). Furthermore, it is possible that this high flux imposes roles for Ist1 in ESCRT-III assembly/function beyond what could be measured previously in other systems at steady state. Nevertheless, there are numerous examples in which endocytic flux is elevated during the development of multicellular organisms. Our work capitalizes on one of these periods and identifies a previously unappreciated role for Ist1 in ESCRT-III assembly and function.

With regard to a role for Ist1 in endocytic recycling, our data are inconsistent with prior findings in the literature. However, we cannot (and do not) state that prior work done in vitro (e.g., using tissue culture cells) was an artefact.

Reviewer #3: The current observations support the conclusion that ESCRT-I and -II do not play a role in membrane deformation. While this conclusion is not surprising to those that have followed the field, it is unclear whether unique aspects of this experimental system impact the generality of this conclusion.

We agree with the reviewer that our findings support the conclusion that ESCRT-I/II do not play a role in membrane deformation in vivo. Given that many of our conclusions, which were previously based only on study of the oocyte-to-embryo transition, have now been shown to apply in another tissue, we believe that our findings regarding ESCRT-III function are likely to be more general.

Reviewer #3: The observation that multiple ILVs have contact with one another may suggest that multiple vesicle formation events are linked (without intermittent scission). Whether this is something that occurs during states of high flux or represents a universal feature of ILV biogenesis that has never been observed before is not clear. These studies are suggestive but do not directly address whether stable microdomains of ESCRT function exist.

We now show that tethers are observed between ILVs in another tissue, which does not exhibit high flux through the ESCRT-dependent MVE biogenesis pathway, suggesting that this is a universal feature of ILV biogenesis (please see revised Figs. 2 and 4). We agree that our studies are only suggestive that stable subdomains of ESCRT function exist, and we have toned down the text regarding this idea in the manuscript (please see page 7).

Reviewer #3: Demonstration that Ist1 can function upstream of Vps2 in sorting events during oocyte-to-embryo transition needs to compare Vps32 localization in contexts where Vps4 function is perturbed. A possible explanation for failure of Vps32 to accumulate to the same degree in ist1 vs vps2 is that vps2 is defective for ESCRT-III disassembly, while Ist1 may not be (based on previous data). If IST1 is epistatic to VPS2 then ist1 and ist1 vps2 morphologies would be indistinguishable.

Unfortunately, as we note in our manuscript, penetrant depletion of Vps4 results in sterility (i.e., no embryos are produced). We attempted to conduct partial depletions, but it was not feasible to know whether a sufficient level of depletion was achieved in order to appropriately interpret the results. We agree that there may exist other explanations for

our data, which we now discuss in the text (please see page 15). Nonetheless, our data indicate that inhibition of Ist1 reduces Vps32 accumulation as compared to control (as well as in comparison to Vps2 depletion). Importantly, inhibition of both Ist1 and Vps2 mimics the effect of Ist1 inhibition alone, consistent with the idea that Ist1 is epistatic to Vps2.

Reviewers' Comments:

Reviewer #1:

Remarks to the Author:

Fine with me now.

Reviewer #2:

Remarks to the Author:

Most of my concerns have been well addressed.

I still have concerns with the wording of these statements:

(1) line 235: "Although upstream ESCRT complexes are required to recruit and promote the assembly of ESCRT-III at MVEs, our data contrast prior in vitro studies, which argued that ESCRT-I and ESCRT-II play an active role in the process of membrane bending¹⁸."

I still don't see how showing that ESCRT-III is required for ILV formation precludes an additional requirement for ESCRT-I and II in active membrane bending. Wouldn't you get the same results after ESCRT-III depletion, if ESCRT-III is required for bending, whether or not ESCRT-I/II also contribute actively to membrane bending, if ESCRT-I/II were not sufficient to bend the membranes without ESCRT-III? The current wording makes it sound as if the results presented preclude an additional role for ESCRT-I and II in membrane bending.

(2) line 231: "Together, these data suggest that proper ESCRT-III assembly is required to maintain cargo clustering independently of ESCRT-0 subdomain formation."

and

line 350: "Moreover, we show that this action occurs independently of ESCRT-0 subdomain formation, which fail to correspond to the precise sites of membrane budding or ESCRT-III assembly in vivo⁶⁷."

I believe that these statements mainly refer to the experiment where, upon Vps32 knockdown, the size of Hrs subdomains increases but Cav1 clustering does not also increase. While this result is suggestive, it's not clear that Cav1 can be clustered more than observed in WT, so this result does not say much about the role of ESCRT-0 in clustering of cargo. The above statements should be toned down with respect to ESCRT-0. One would need to assay cluster formation after ESCRT-0 knockdown, and observe normal clustering, to make a strong case for clustering being ESCRT-0 independent.

Reviewer #3:

Remarks to the Author:

In this revised manuscript the authors have gone to some lengths to address concerns of the reviewers. This remains a very nice piece of work that takes aim at several "controversies" within the field, often times with surprising results. While this is interesting it remains difficult to appreciate whether the differing results are a consequence of the very unique system employed, in which case they may reveal variations on a theme and even suggest that laws may not apply to ESCRT function from cell type to cell type. More than this breaks new ground, it revisits concepts that have been around as long as the ESCRT complexes themselves – some of the data supports those models, other does not. That is fine, but it would be beneficial to represent the work in this frame.

This work does not "define mechanisms" as asserted in the Abstract, but it does throw its weight

behind existing models. That is fine, but it is important to deliver on what is promised, reflect upon what the data really indicates, and put data in the context of previous work.

The introduction states that Ist1 has been shown to positively impact ESCRT function. This is true, but it has also been demonstrated to have a negative effect on ESCRT function, at least in part via direct interaction with Vps4. This seems particularly relevant in the present studies. If at the start of these studies "the role of Ist1 in endosomal protein sorting (was) elusive" how has it been clarified through these studies? While the authors have documented some phenotypes for Ist1 perturbation it isn't clear that these studies have done anything more than confirm or refute previous findings. Because of the quality of the present studies this isn't a fatal flaw, but the limitations should be acknowledged. For instance, it is possible that phenotypes observed upon IST1 deletion are actually resulting from synthetic genetic interactions that result from stage-specific expression (in this case lack thereof). That doesn't change the observation, merely the conclusion, which in this case is being made forcefully. (These are essentially the same concerns voiced in the first round.) The Results section highlights how unique this system is and why we should want to read this work, but the fact that it is so unique tempers the ability to make statements chiseled in stone. Please acknowledge this caveat when making bold claims.

In the Introduction: "we demonstrate that ESCRT-III is directly responsible for membrane bending in vivo." While it is possible to demonstrate a requirement for ESCRT-III in these events it is virtually impossible to prove that this is direct in vivo.

Manuscript # NCOMMS-17-07153A; Revision submission date: September 7, 2017
Title: "Ist1 regulates ESCRT-III assembly and function during multivesicular endosome biogenesis"

Response to Referee comments:

Reviewer #1: Fine with me now.

We are pleased that the reviewer does not have any further concerns.

Reviewer #2: Most of my concerns have been well addressed.

I still have concerns with the wording of these statements: (1) line 235: "Although upstream ESCRT complexes are required to recruit and promote the assembly of ESCRT-III at MVEs, our data contrast prior in vitro studies, which argued that ESCRT-I and ESCRT-II play an active role in the process of membrane bending¹⁸." I still don't see how showing that ESCRT-III is required for ILV formation precludes an additional requirement for ESCRT-I and II in active membrane bending. Wouldn't you get the same results after ESCRT-III depletion, if ESCRT-III is required for bending, whether or not ESCRT-I/II also contribute actively to membrane bending, if ESCRT-I/II were not sufficient to bend the membranes without ESCRT-III? The current wording makes it sound as if the results presented preclude an additional role for ESCRT-I and II in membrane bending.

We are pleased that most of the reviewer's concerns were addressed. With regard to line 235, we have modified the text to be clearer (page 10). While we agree that our data do not preclude an additional role for ESCRT-I and ESCRT-II in membrane bending, the previous work cited on line 235 suggested that ESCRT-III is not required for membrane bending. Our findings contradict this idea, and instead argue that ESCRT-III plays an important role in membrane bending.

Reviewer #2: (2) line 231: "Together, these data suggest that proper ESCRT-III assembly is required to maintain cargo clustering independently of ESCRT-0 subdomain formation." and line 350: "Moreover, we show that this action occurs independently of ESCRT-0 subdomain formation, which fail to correspond to the precise sites of membrane budding or ESCRT-III assembly in vivo⁶⁷." I believe that these statements mainly refer to the experiment where, upon Vps32 knockdown, the size of Hrs subdomains increases but Cav1 clustering does not also increase. While this result is suggestive, it's not clear that Cav1 can be clustered more than observed in WT, so this result does not say much about the role of ESCRT-0 in clustering of cargo. The above statements should be toned down with respect to ESCRT-0. One would need to assay cluster formation after ESCRT-0 knockdown, and observe normal clustering, to make a strong case for clustering being ESCRT-0 independent.

We appreciate the reviewer's comments, and we have toned down the text with regard to ESCRT-0 (pages 9 and 14). It is important to note, however, that we also demonstrate that the loss of Ist1 has a dramatic impact on cargo distribution, which supports the idea that ESCRT-III assembly is necessary to maintain cargo clustering at MVEs.

Reviewer #3: In this revised manuscript the authors have gone to some lengths to address concerns of the reviewers. This remains a very nice piece of work that takes aim at several "controversies" within the field, often times with surprising results. While this is interesting it remains difficult to appreciate whether the differing results are a consequence of the very

unique system employed, in which case they may reveal variations on a theme and even suggest that laws may not apply to ESCRT function from cell type to cell type. More than this breaks new ground, it revisits concepts that have been around as long as the ESCRT complexes themselves – some of the data supports those models, other does not. That is fine, but it would be beneficial to represent the work in this frame. This work does not “define mechanisms” as asserted in the Abstract, but it does throw its weight behind existing models. That is fine, but it is important to deliver on what is promised, reflect upon what the data really indicates, and put data in the context of previous work.

We have adjusted the wording in the abstract. We believe that our studies make an important contribution to the overall understanding of how MVE formation is regulated by the ESCRT machinery, irrespective of the model system chosen for analysis.

Reviewer #3: The introduction states that Ist1 has been shown to positively impact ESCRT function. This is true, but it has also been demonstrated to have a negative effect on ESCRT function, at least in part via direct interaction with Vps4. This seems particularly relevant in the present studies. If at the start of these studies “the role of Ist1 in endosomal protein sorting (was) elusive” how has it been clarified through these studies? While the authors have documented some phenotypes for Ist1 perturbation it isn’t clear that these studies have done anything more than confirm or refute previous findings. Because of the quality of the present studies this isn’t a fatal flaw, but the limitations should be acknowledged. For instance, it is possible that phenotypes observed upon IST1 deletion are actually resulting from synthetic genetic interactions that result from stage-specific expression (in this case lack thereof). That doesn’t change the observation, merely the conclusion, which in this case is being made forcefully. (These are essentially the same concerns voiced in the first round.) The Results section highlights how unique this system is and why we should want to read this work, but the fact that it is so unique tempers the ability to make statements chiseled in stone. Please acknowledge this caveat when making bold claims.

We believe that our findings provide the first convincing evidence demonstrating that Ist1 plays a key role during ILV formation at MVEs. Throughout the manuscript, we repeatedly temper our conclusions, emphasizing that our findings are made under conditions of high endocytic flux and an elevated rate of MVE formation. However, it is also important to highlight that all of our studies focus on a native developmental timepoint in a model system that has contributed extensively to our understanding of cell biology in general. Moreover, we believe it is essential to explore the function of the ESCRT machinery during early embryogenesis, an underexplored phase of development, which cannot be addressed using yeast or mammalian tissue culture cells (the primary systems currently under scrutiny). Nonetheless, in this revision, we have adjusted the title to emphasize again that our work was conducted using *C. elegans*.

Reviewer #3: In the Introduction: “we demonstrate that ESCRT-III is directly responsible for membrane bending in vivo.” While it is possible to demonstrate a requirement for ESCRT-III in these events it is virtually impossible to prove that this is direct in vivo.

We have adjusted the text on page 4 to address this concern.